# Use of Stability Modeling to Support Accelerated Vaccine Development and Supply

**DOI:** 10.3390/vaccines9101114

**Published:** 2021-09-30

**Authors:** Cristiana Campa, Thierry Pronce, Marilena Paludi, Jos Weusten, Laura Conway, James Savery, Christine Richards, Didier Clénet

**Affiliations:** 1GSK, Technical R&D, 53100 Siena, Italy; cristiana.x.campa@gsk.com (C.C.); marilena.x.paludi@gsk.com (M.P.); 2GSK, Technical R&D, 1330 Rixensart, Belgium; thierry.x.pronce@gsk.com; 3MSD, Center for Mathematical Sciences, 5344 Oss, The Netherlands; jos.weusten@merck.com; 4Merck, Regulatory Affairs CMC Vaccines, North Wales, PA 19454, USA; laura_conway@merck.com; 5AstraZeneca, Data Science & Modeling, BioPharmaceuticals Development, R&D, Cambridge 01223, UK; james.savery@astrazeneca.com; 6Sanofi-Pasteur, Global Quality, Swiftwater, PA 18370, USA; Christine.L.Richards@sanofi.com; 7Sanofi-Pasteur, Bioprocess R&D Department, 69280 Marcy l’Etoile, France

**Keywords:** accelerated predictive stability, advanced kinetic modeling, COVAX

## Abstract

Stability assessment of pharmaceuticals in specific storage and shipment conditions is a key requirement to ensure that safe and efficacious products are administered to patients. This is particularly relevant for vaccines, with numerous vaccines strictly requiring cold storage to remain stable. When stability evaluation is exclusively based on real-time data, it may represent a bottleneck for rapid and effective vaccine access. Stability modeling for vaccines represents a key resource to predict stability based on accelerated stability studies; nevertheless, this approach is not fully exploited for these kinds of products. This is likely because of the complexity and diversity of vaccines, as well as the limited availability of dedicated guidelines or illustrative case studies. This article reports a cross-company perspective on stability modeling for vaccines. Several examples, based on the direct experience of the contributors, demonstrate that modeling approaches can be highly valuable to predict vaccines’ shelf life and behavior during shipment or manipulation. It is demonstrated that modeling methodologies need to be tailored to the nature of the vaccine, the available prior knowledge, and the monitored attributes. Considering that the well-established strategies reported in ICH or WHO guidelines are not always broadly applicable to vaccines, this article represents an important source of information for vaccine researchers and manufacturers, setting the grounds for further discussion within the vaccine industry and with regulators.

## 1. Introduction

To successfully carry out global immunization programs, most vaccines must be kept refrigerated or frozen with the aim of ensuring ensure their stability from production to use. The World Health Organization (WHO) estimated that cold chain breaks (i.e., excessive temperature excursions outside of the recommended storage conditions) are responsible for around 50% of vaccine wastage [1], pointing out how the temperature is one of the key parameters impacting the stability of vaccines. The current regulatory framework of biological pharmaceuticals’ stability is governed by several interrelated guidance documents. In North America, the European Union, Japan, and China, these guidance documents are provided through the International Conference on Harmonization (ICH) of Technical Requirements for Registration of Pharmaceuticals for Human Use. The design and analysis of stability studies are outlined in the ICH Q1 series, complemented, for biologicals, by the ICH Q5C. Additional guidance can also be found in the WHO/BS/06.2049 (Guideline on stability evaluation of vaccines).

According to those sources, primary data to support a requested storage period (expiry) for either a drug substance or drug product should be based on long-term, real-time, real-condition stability studies, in support of INDs/IMPDs or marketing applications. For initial licensing, both the drug substance and drug product require a minimum of three batches with a minimum of 6 months’ stability data at the time of submission to be submitted when requesting an expiry greater than 6 months. The use of data generated under stress/accelerated conditions is very limited in the overall analysis and, even though the ICH Q5C does point out the use of accelerated data, it does so in a very generic sense:

“*Studies under accelerated conditions may provide useful support data for establishing the expiration date, provide product stability information for future product development (e.g., preliminary assessment of proposed manufacturing changes such as change in formulation, scale-up), assist in validation of analytical methods for the stability program, or generate information which may help elucidate the degradation profile of the drug substance or drug product. Studies under stress conditions may be useful in determining whether accidental exposures to conditions other than those proposed (e.g., during transportation) are deleterious to the product and also for evaluating which specific test parameters may be the best indicators of product stability*.”

However, accelerated storage conditions should also be taken into consideration not only when establishing expiration dating, but also to factor in distribution and end-user times and conditions, as they provide insightful information into the maintenance of molecular conformation, biological activity, and understanding of degradation pathways.

The current regulatory framework also does not propose any specific approaches/statistical models for the extrapolation of shelf-life for biologics, thus referring back to the approaches laid out in the ICH Q1E for data interpretation and modeling. However, the approaches therein refer back to simple linear regression and associated confidence intervals and may not be suited to the kinetics observed in complex biological products. The ICH Q1E Guideline also mentions that, “in some cases, a non-linear regression can better reflect the true relationship”; however, no explanation on the methodology is reported.

Vaccine manufacturing and testing require a significant period of time, and vaccine shelf-life can be set at up to 3 or 4 years to ensure enough residual shelf-life for the patients.

Therefore, the evaluation of the stability profile to set the expiry of complex biologicals or vaccines will continue to be on the critical path for development and availability of new vaccine candidates and shelf-life extension during the vaccine life cycle, if solely based on experimental data (real-time/temperature excursion studies until shelf-life). This rigid methodology is not compatible with the accelerated vaccine development and industrial plan needed to address a pandemic situation such as the current COVID-19 emergency or the efficient extension of shelf-life.

A potential solution to this situation would be to rely more on a holistic approach to data trending and analysis for shelf-life prediction, outside of the current, very limited scope allowed in ICH Q1A and bound to the notion of “significant change” specific to the following: (A) drug substance, (1) as failure to meet specification; (B) drug product, (1) a 5% change in assay from its initial value, or failure to meet the acceptance criteria for potency when using biological or immunological procedures; (2) any product degradant exceeding its acceptance criterion; and (3) failure to meet the acceptance criteria for appearance, physical attributes, and functionality test (e.g., color, phase separation, resuspendability, caking, hardness, dose delivery per actuation); however, some changes in physical attributes may be expected under accelerated conditions.

While those criteria may seem to be transposable to biologics, they do not necessarily reflect the complexity and variability of the assays used for monitoring the stability profiles of those products, and alternative approaches would be desirable. A solution to this situation would be to allow the use of new modeling approaches, utilizing the Arrhenius equation, advanced kinetics, or more generally advanced statistical analysis of stability data coming from stress/accelerated conditions during development to ensure (1) prediction of shelf-life; (2) management of temperature excursions (cold chain breaks); (3) understanding the impact of degradation on product quality; and (4) management of post-approval changes and comparability studies following a manufacturing change.

In this paper, we will give a short description of the theory behind the advanced kinetic models together with potential experimental considerations. Subsequently, the concepts will be illustrated using several real-life examples with varying degrees of model complexity. The examples illustrate the usefulness of data obtained at elevated temperatures to accelerate degradation in the assessment of the kinetic parameters such as long-term stability predictions. It will also be shown that the models can adequately predict degradation under fluctuating temperature conditions, for example, during shipment. Finally, it will address why the current ICH approach to stability assessment should be widened to include the use of advanced kinetic models.

## 2. Materials and Methods

### 2.1. Modeling Approaches

Though linear regression models can be useful tools to describe the degradation of the stability measure over time, it is to be acknowledged that many degradation processes do not follow such a simple profile. Especially in accelerated studies, using storage temperatures exceeding 30 °C, the profiles will generally be markedly non-linear. Moreover, the linear models obtained at given storage temperatures do not provide a direct tool to predict the rate of degradation at other temperatures or under varying temperatures. In these situations, more sophisticated models can be used, capturing the degradation at multiple temperatures in a single model [2]. By applying so-called ‘advanced kinetic modeling’ to multiple observations obtained over a relatively short period of time at multiple temperatures, successful predictions can be made of the degradation over prolonged periods of time, which is especially useful in the case of a pandemic when the world simply cannot wait for real-time stability data over multiple years to be generated.

### 2.2. Theoretical Background

It is useful to go into some detail concerning the concepts underlying this advanced kinetic modeling. From a chemical point of view, degradation of active components in vaccines is a complicated process, and will depend on the exact composition of the vaccine. It is possible that the active component A disintegrates to components B and C via a simple reaction (A → B + C), for example, owing to oxidation or hydrolysis. It is also readily possible that there are multiple reactions (A → B + C and A → D + E), or that product B acts as a catalyst to speed up the degradation (A + B → 2B + C), and so on. The reaction rates are also temperature-dependent. In spite of the diversity and complexity, it appears possible to capture many of these processes in a relatively simple equation. At the heart of the kinetic model is a differential equation describing the propagation of the reaction over time, given by the symbol *α*. If no conversion has taken place, then *α* = 0, and if the reaction has proceeded completely to its end, it follows *α* = 1. Mathematically, the rate of change in *α* with time t is depicted by ∂α/∂t. The so-called truncated Šesták–Berggren approach [2] states that the reaction rate of many reactions can be captured in
(1)$$\frac{\partial \alpha }{\partial t\;}=k{\left(1-\alpha \right)}^{n}\alpha^{m}$$
for some values of the kinetic parameters, namely *k*, the rate constant; *n*, the order of the reaction; and m, a parameter introduced to take into account the possible autocatalytic behavior of reaction (1). If two reactions play a role, this is expanded to
(2)$$\frac{\partial \alpha }{\partial t}=k_{1}{\left(1-\alpha \right)}^{n_{1}}\alpha^{m_{1}}+k_{2}{\left(1-\alpha \right)}^{n_{2}}\alpha^{m_{2}}$$

In case of the simple reaction A → B + C, a model of Equation (1) is obtained with *m* = 0. Simple linear and first-order kinetics assume *n* = 0 and *n* = 1, respectively, while higher values are required for higher-order kinetics.

A first order reaction A → B + C with an autocatalytic reaction A + B → 2B + C is typically described as a one-step reaction by the model of Equation (1) with *m* ≠ 0.

Reviews of the underlying theory can be found in, for example, [2,3,4].

The factor *k* in Equations (1) and (2) is a temperature-dependent reaction rate constant, generally assumed to follow the Arrhenius equation
(3)$$k=A\;Exp\left[-\frac{E_{a}}{R\;T}\right]$$with *A* being the pre-exponential factor, *E_a_* being the activation energy in J·mol^−1^, *R* being the universal gas constant (*R* = 8.314 J·mol^−1^·K^−1^), and *T* being the temperature in Kelvin. The factor *A* accounts for the probability that molecules that should react collide with sufficient force and in the proper conformation to allow a reaction, and *E_a_* is the amount of energy that has to be overcome for the reaction to proceed; generally, reactions proceed via unstable intermediates requiring energy to be formed.

The two-step differential equation as described in Equation (2) is generally sufficient to adequately describe all relevant kinetic models, from the simplest to the more complex [2]. Two-step profiles are often mentioned to describe complex bioproduct degradations with an initial rapid drop followed by a long gradual decrease phase. This is especially the case for viruses such as measles [5], a vero cell-adapted rinderpest vaccine [6], canine distemper poultry infectious bronchitis viral vaccines [7], Rabies [3], Polio, Yellow fever [8], and ALVAC poxvirus [9]. These models not only provide an adequate description of the data at hand, but can also be used to predict values after prolonged times. As the model includes different temperatures, it is also possible to describe and predict profiles with non-constant temperatures. For example, release limits can be set by backwards reasoning from a desired expiry date assuming storage for well-defined time frames at 25 °C, 15 °C, and 5 °C. A more complex situation is real-life transport, during which the temperature may not be constant (see examples in Section 3.5, below). Estimation of shelf life and recommendations concerning storage temperatures are among other possible applications of the model. Owing to the very different chemistry and physics for frozen products compared with products in the liquid state, the models are used for their own physical state, leading to different kinetic parameters for the liquid or the solid state. It is important to stress that, although there is a basis in the theory on chemical reaction kinetics, the equations represent an empirical model.

### 2.3. Experimental Considerations

Building a kinetic model from experimental data means the determination of the rate of degradation of a product as a function of time and temperature. Stability indicating key attributes first have to be identified for the product of interest. It can be infectious titer determined by plate assay or CCID50 for live-attenuated vaccines, antigenicity determined by ELISA for inactivated virus-based vaccines, the proportion of unexpected oligomeric forms by HP-SEC for sub-unit vaccines, and so on. Using dedicated software, it is possible to estimate the parameter values according to different models (e.g., a single-step model (Equation (1)) versus a two-step model (Equation (2))) and to apply statistical criteria to select the so-called ‘best’ model, giving an adequate description of the data with the lowest number of parameters. Because, in many cases, it appears that a one-step model is not enough to adequately describe the data, a two-step model is often needed, with several parameters to be estimated from the data. For a suitable analysis, it is required that the relevant phenomena are captured in the data; if there is an initial rapid drop, then observations should be made during the appropriate time frame. It is further important that the experimental data cover multiple temperatures (e.g., 5 °C, 25 °C, and 37 °C and/or 40 °C) and that the results do show a clear time effect in the observed quality attributes; if the totality of data indicate almost perfect stability over all time frames at all temperatures, there is nothing to model. Experience so far indicates that more than 20% degradation should be observed under the more aggressive conditions (e.g., weeks or months at 40 °C), and that about 20–30 data points are recommended to allow reliable estimation, including replicated observations at individual time points. Furthermore, the use of a methodology of model selection such as Akaike and Bayesian information theory is strongly recommended to avoid overfitting of experimental data and select the more appropriate kinetics models. These practical considerations have to be considered as good modeling practices [4,10].

## 3. Results

### 3.1. First-Order Kinetic Model Enabled Long-Term Stability Predictions of a Glycoconjugate Vaccine at +5 °C

O-acetylation (OAc) content is one of the relevant quality attributes for GSK‘s conjugate vaccines’ immunogenicity, particularly for the meningococcal serogroup A component. A rate loss of O-Acetyl groups from the conjugate MenA component is expected over time and temperature, especially when the vaccine is presented in the liquid state. In this example, a first-order kinetics was identified to best describe the chemical reaction of loss of the O-acetyl moiety of a vaccine. Note that Equation (1) with n = 1 and m = 0 implies that the degradation can be described by a straight line when plotting the response on a logarithmic scale. During the development of a liquid formulation, stability data were collected on MenA polysaccharide stored at four temperatures: +15 °C, +25 °C, +37 °C, and +50 °C for a period of up to 16 weeks. The linear fit of experimental data presented in Figure 1a indicates that the first-order kinetics may be used here for describing the rate of chemical reaction of loss of the O-acetyl moiety. Arrhenius-based kinetic modelling was used to design the aging process for the investigational tailored vaccines with a predefined level of OAc MenA attribute to obtain the proven clinical specifications [11].

The model built on the MenA polysaccharide process intermediate could be used to predict the stability profile for a final product vaccine lot when stored at the recommended temperature of +5 °C, for which only the release data are known. Simulation techniques could be applied to embed analytical variability into the model and determine a range of possible future results.

A graphical example is presented in Figure 1c. The filled circle at time 0 represents the level of the attribute in the vaccine lot at release date and is used together with Arrhenius-based kinetic modelling to build the stability model at +5 °C (solid line). The dashed lines are the prediction bands at 95% of coverage for the vaccine lot. Empty circles depict observed results that were not used in the fit during kinetic analysis, allowing verification of the predictions. The presented results reveal that, for a chemical reaction like the loss of O-Acetyl moiety, good predictions over more than 3 years can be made using the Arrhenius-based kinetic model based on only 16 weeks of stability data.

### 3.2. n^th^-Order Kinetic Model Enabled Accurate Long-Term Stability Predictions of a Protein-Based Vaccine at +5 °C and +25 °C

In this example, a Sanofi protein-based vaccine consisting of the antigen PhtD adsorbed to aluminum salts is considered. A set of experimental data was determined over 6 months by RP-HPLC and contained the dependence of concentration of intact protein as a function of time under recommended (+5 °C) and elevated (+25 °C, +37 °C, and +45 °C) storage temperatures [9]. The use of empirical models (zero- and first-order reactions (Equation (1) with m = 0 and n = 0 or n = 1, respectively; dotted and dashed lines in Figure 2b, respectively) led to an agreement between predictions and experimental data at +5 °C, but not at +25 °C. A more appropriate n^th^-order model (solid lines, Figure 2), identified as the so-called ‘best’ model by applying statistical comparison criteria, was able to accurately predict long-term reaction progress at both +5 °C and +25 °C (Figure 2b).

Using this best kinetic model and applying bootstrap analysis, 95% prediction bands were constructed for the +5 °C and +25 °C conditions, and the long-term experimental data (open circles in panels c and d of Figure 2) were found to be well in line with the prediction.

### 3.3. Using n^th^-Order Kinetic Models for Long-Term Stability Predictions at +5 °C of Multiple Batches of a Vaccine

The quality parameter for MSD’s vaccine is the antigen content as estimated via an ELISA method. Data were available for 19 batches, up to 6 months at +25 °C and +37 °C, and up to 2.5 years at +5 °C.

The data were fitted using the one-step kinetic model (Equation (1)), with m = 0. Only the data up to 6 months were used to build the models, allowing comparison of the observed values with the predictions at 5 °C for prolonged times. The results of the batches were found to be consistent in the sense that the data of all 19 batches could be fitted with a single set of kinetic parameter values (the parameters A and E_a_ of Equation (3), and n of Equation (1)).

The results are presented graphically in Figure 3. Each panel represents a different batch. The dashed lines represent the two-sided 95% prediction intervals for the 5 °C condition. Open circles present observed results that were not used in the fit, allowing verification of the predictions. The results reveal that predictive bands contained most of the experimental data over the full 2.5 years using the kinetic model when using data up to 6 months only.

### 3.4. Two-Step Kinetic Model Required for Accurate Long-Term Stability Predictions of an Inactivated Virus-Based Vaccine at +5 °C

In this example, an inactivated virus-based vaccine containing several variants and formulated as a freeze-dried product at Sanofi was used to compare prediction of antigenicity and long-term experimental stability data [3]. Using the classical linear fit of short-term experimental data at +5 °C is not appropriate to predict long-term stability at +5 °C (see Figure 4a,c). Similarly, a lack of accurate long-term prediction at +5 °C was observed using empirical one-step zero- or first-order reactions (not shown). In such cases, more sophisticated kinetic equations (two-step models; Equation (2)) were required, enabling accurate long-term stability predictions, as selected by an Aaike and Bayesian statistical scoring (AIC/BIC). Similarly to the previous example, this case illustrates the usefulness of the use of stability data obtained not only at +5 °C, but also at higher incubation temperatures that contain key information.

In this situation, considering 60% of antigenicity recovery as the lower acceptance level for this experimental vaccine and the lower predictive band, vaccine stability for at least 3 years at +5 °C can be claimed only using the best kinetic model (Figure 4, panel d), whereas the use of a poor model based on limited information (i.e., only +5 °C data) unrealistically shortens the proposed shelf-life (Figure 4, panel c).

### 3.5. Temperature Excursions’ Management and Real-Time Stability Monitoring of Vaccines

The examples shown in the previous sections all concern stability studies at constant temperatures. The kinetic model, however, can also be applied under non-stable temperatures, by solving the differential equations numerically given the temperature profile over time after the values of the kinetic parameters have been estimated. This is important for vaccines, especially for thermosensitive ones recently developed against COVID-19, as there is no guarantee that, during transport, for example, the temperature will remain constant and, at the point of care where the vaccine is administered, the conditions may not be as tightly controlled as in the labs of the manufacturer [12,13,14].

The examples in Figure 5 show real-life examples under fluctuating temperatures. Given the release value, the kinetic parameters, and the temperature profile, the time profile of the loss of antigenicity was predicted. The impact of the variable temperature on the degradation profile is clearly visible in the plots. At the end of the time frame, the actual antigenicity was measured and compared with the prediction, with actual data points falling within the prediction intervals, confirming the accuracy of the predictive models. Furthermore, the impact of successive excursions at ambient temperature experienced by a vaccine can easily be estimated. For the vaccine presented in Figure 5b, six short excursions around +22 °C did not have a significant impact on its antigenicity. These examples illustrate the power of the kinetic models in ensuring the quality of the products throughout their existence, from production to use.

## 4. Discussion

The results presented in this manuscript highlight how accurate stability predictions of vaccines can be performed during their storage and shipments as appropriate kinetic models are developed. Over the last years, discussion has been progressing between industry and some regulatory agencies on the use of stability modeling to support biologics development and shelf life assignment, especially in the case of unmet medical need. For instance, during the EMA/FDA stakeholder workshop on support to quality development in early access approaches (2018), the use of stability modeling grounded in prior knowledge was discussed for monoclonal antibodies [15]. These reflections set the basis for the recent “EMA Draft toolbox guidance on scientific elements and regulatory tools to support quality data packages for PRIME marketing authorization applications” [16], which also includes a section dedicated to stability predictions for biotech products. More technical information on the proposed approach, mostly applied for monoclonal antibodies, is reported in [17].

For complex bioproducts such as vaccines, stability prediction requires specific considerations, owing to the diversity of platforms and the variety of stability indicating properties. As demonstrated in this paper (and the relevant references therein), tailored modeling approaches are needed, depending on the monitored attributes and degradation pathways. Nevertheless, for a given vaccine platform (e.g., mRNA, viral vectors), prior knowledge elements could be considered for stability predictions. Thanks to the prior knowledge approach, appropriate accelerated stability studies including relevant attributes and time points/temperature of interest can easily be designed.

Additionally, more accurate designs of stability studies leading to a reduction in required stability data could be expected. The use of prior knowledge combined with kinetic-based modeling approaches represents a robust approach for products’ stability assessment [4,17]. Using good modeling practices, the fitting of experimental stability data obtained at different temperatures led to the identification of models that best describe the change in key stability attributes as a function of time. Depending on the considered attribute and the complexity of degradation occurring in products, simple to more sophisticated models are selected, leading to the use of well-fit kinetic equations [2]. A simple model such as first-order kinetics with single-step activation provided a consistent framework to describe the loss of infectivity for viral vectors or adenoviruses [18,19], whereas more complex two-step models were required to accurately predict virus-based vaccines [2,4]. Such advanced-kinetic modeling makes it possible to go beyond the ICH and WHO recommendations for stability predictions of products [20]. Furthermore, such models can also be used to adequately predict degradation of products in real time under standard storage conditions (i.e., 2–8 °C) and under fluctuating temperature conditions (cold-chain breaks). Beyond the vaccine vial monitor (VVM) [13,21,22] and extended controlled temperature chain (ECTC) [23] initiative of the WHO, the integration of kinetic modeling in supply chain product management could dramatically improve the monitoring of the quality of vaccines during their shipping and use, averting product wastage, even after experiencing minor excursions [24]. Kinetic-based modeling approaches presented in this paper are based on sound scientific knowledge, leading to better stability prediction accuracy than classical approaches based on mean kinetic temperature (MKT), as proposed in USP<1079.2> for example, which is not appropriate for use in a transportation risk assessment [24,25].

Another important point to consider when using modeling approaches for stability prediction is to ensure batch-to-batch consistency of the predictive model. Starting from the current ICH Q1 recommendation for data at filing, a minimum of three batches should be considered for the initial model establishment. The model should be further monitored and updated as new data become available in a continued process verification (CPV)-like approach, and any data point falling outside of the prediction interval is duly investigated as being out-of-trend (OOT). Validity and stability reliability over time of the model prediction will thus be strengthened as the core data expand, and any potential drifts are captured and assessed. Extending that approach, and based on the modelling purposes, models could be constructed for a single batch, for a dedicated product (intermediate, DS or DP), or even for a dedicated product range, should the behavior of the stability profiles be similar. The latter could cover, for example, multiple primary packaging of the same material or a dedicated manufacturing platform (mRNA vaccines), leading to the need to establish statistical approaches for the poolability of data. Although the general approaches described in ICH Q1E (Section B.2.2; ANCOVA—analysis of covariance) could be applied to the simpler model (linear-Arrhenius), more complex ones will have to be devised for higher-order and other, more complex kinetics. Normalization of the data may also have to be considered if the manufacturing variability is too high and leads to a wide distribution of data at time zero (batch manufacture).

The emergency posed by COVID-19 has further fostered discussions on this topic, especially for vaccines. For instance, a COVAX [26] workshop dedicated to stability strategies was recently held, highlighting the importance of using stability models for COVID-19 vaccines development and supply [27]. In the WHO document entitled “Considerations for evaluation of COVID-19 vaccines—Points to consider for manufacturers of COVID-19 vaccines” (25 November 2020) [28], it is mentioned that, “*generally, real-time stability from three full-scale production lots is preferred. With appropriate justification and discussion with the WHO, a scientific risk-based approach to determine the proposed vaccine shelf life in the absence of real time stability data on the commercial batches may be considered. For example, data generated from smaller lots, such as clinical or engineering lots, and/or data generated on a different vaccine using a similar process and/or manufacturing platform, may be appropriate for submission in support of the initial recommended shelf-life for the vaccine. Consideration of platform stability data, prior knowledge from early clinical batches or statistical modelling may also be applied to forecast expiry of product*”. Stability modeling may also help with rapid introduction of vaccines for COVID-19 variants. For instance, in the EMA “Reflection paper on the regulatory requirements for vaccines intended to provide protection against variant strain(s) of SARS-CoV-2” [15], it is mentioned that, “ *in principle, the registered shelf life conditions/period would be applicable. However, confirmation of the suitability of the active substance and finished product registered shelf life needs to be demonstrated (e.g., by available real-time stability data, predictive stability models, early stability data under accelerated storage conditions). Confirmatory real-time stability data need to be provided post-approval*.”

Beyond the emergency, during (accelerated) vaccine development, when knowledge on critical quality attributes and their correlation with protection may evolve, the use of modeling best practices reported in this paper, along with the identification of stability indicating physicochemical or immunochemical properties, is key to support robust product understanding, enabling rapid access to new vaccines without compromising safety and efficacy.

## 5. Conclusions

The recent pandemic period and the urgent need to offer vaccines to people have highlighted the considerable interest in predictive models in order to anticipate the stability of vaccines, especially when only short-term experimental data are available for file submissions. The results presented in this paper illustrate how appropriate kinetic models can be used to accurately predict the long-term stability of complex biological products. Predictive modeling can also be used to monitor in real time the stability of products from their production to their use, as temperature fluctuations experienced by products are known. This enables de-risking of the impact of potential unexpected minor and successive excursions of temperature that can occur during storage or shipments with a solid scientific basis and better accuracy than classical approaches based on mean kinetic temperature (MKT) as proposed in USP. Applied with success for various types of vaccines (recombinant protein sub-unit, viral, glycoconjugates) coming from different companies, such models look reliable and should be used as a routine practice to support accelerated development of vaccines.

## Figures and Tables

**Figure 1 vaccines-09-01114-f001:**
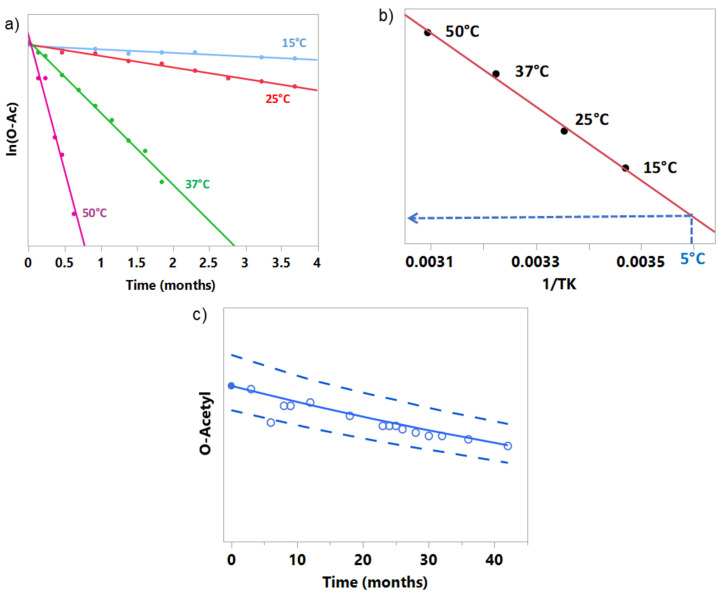
(**a**) Stability data points used to build kinetic models (filled circles) at +15 °C, +25 °C, +37 °C, and +50 °C in panel (**b**) and Arrhenius equation, (**b**) Arrhenius-coordinates plot used for the prediction of the stability at 5 °C (dashed blue lines), and (**c**) prediction of the stability at 5 °C (solid line) with prediction bands at 95% coverage (dashed lines). Filled circle represents the release data, while empty circles represent the experimental stability data.

**Figure 2 vaccines-09-01114-f002:**
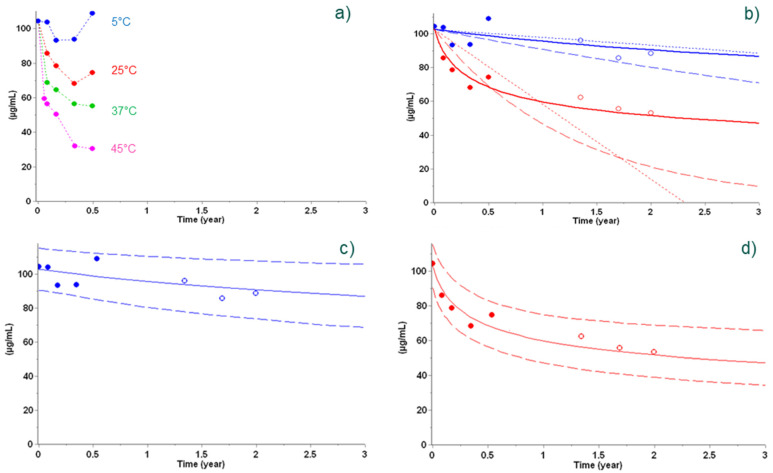
Stability data points used to build models (full circles) at +5 °C (blue), +25 °C (red), +37 °C (green), and +45 °C (pink) (**a**) and not used in the determination of the models (empty circles in panels (**b**–**d**)) overlayed with predictions from empirical models (zero-order in dotted lines and first-order in dashed lines in panel (**b**)) and best kinetic model (n^th^-order reaction in solid lines in panels (**b**–**d**). Prediction using the n^th^-order reaction in panels (**c**,**d**)) is shown with a 95% prediction interval (dashed lines). Reproduced from Clénet et al. (2014), Copyright © 2014, with permission from Elsevier.

**Figure 3 vaccines-09-01114-f003:**
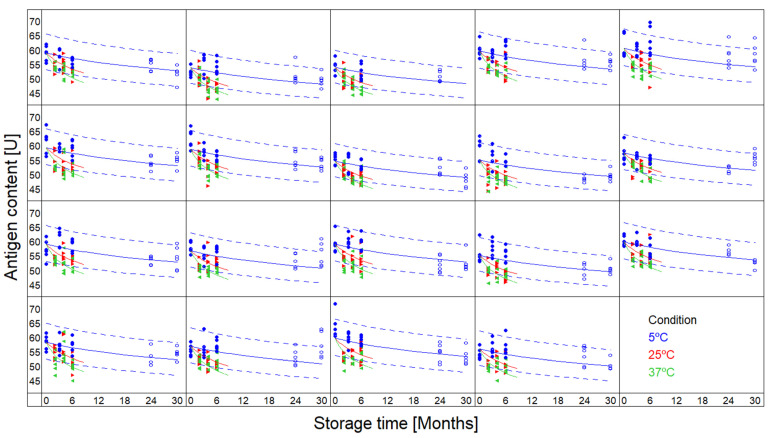
Antigenicity predictions based on vaccine 6 months’ data points (full circles) at +5 °C (blue), +25 °C (red), and +37 °C (green) for 19 batches. Antigenicity predictions at +5 °C are displayed as blue lines based on n^th^-order kinetics best fitting experimental data, including 95% two-sided prediction intervals (dashed-lines). Additional experimental data obtained later in time are displayed as open circles for verification.

**Figure 4 vaccines-09-01114-f004:**
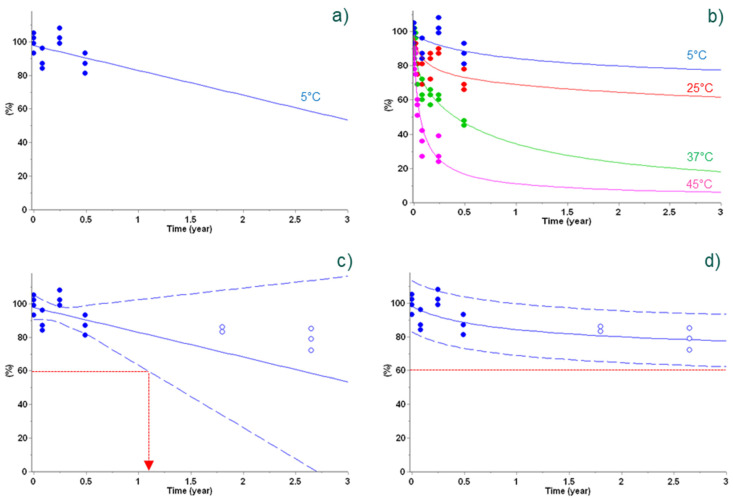
Antigenicity predictions of vaccine using a classical linear regression (left column) and non-linear regression (i.e., advanced kinetics, right column) of 6 months’ data points (full circles). Data at +5 °C (blue) were used for linear regression (panel (**a**)), whereas additional data at +25 °C (red), +37 °C (green), and +45 °C (pink) were also used for non-linear regression (panel (**b**)). Antigenicity predictions are displayed as lines based on the linear model (left column) and best kinetic model (right column), including a 95% two-sided confidence level for the mean for the linear model (dashed lines, panel (**c**)) and 95% bootstrap predictive band for the kinetic model (dashed lines, panel (**d**)). Additional experimental data obtained later in time are displayed as open circles for verification (panels (**c**,**d**)). The lower acceptance limit at 60% is displayed as dotted red lines (panels (**c**,**d**)). Reproduced from Clénet (2018), Copyright © 2018, with permission from Elsevier for right column.

**Figure 5 vaccines-09-01114-f005:**
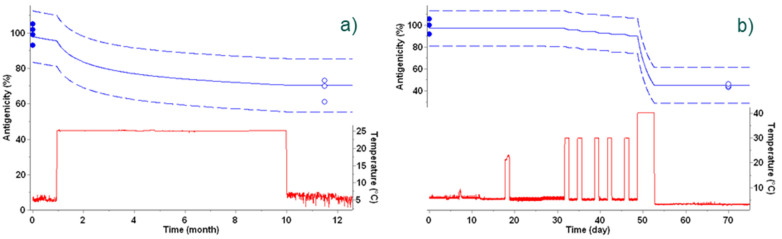
Vaccine antigenicity predicted from the best kinetic model during temperature excursions outside the cold-chain. (**a**) Temperature fluctuation experienced by a vaccine during storage for 12 months including a temperature excursion for 9 months in incubator set at +25 °C is displayed as a red line. Antigenicity predictions are displayed as blue lines, including the 95% prediction band as dashed lines. Experimental data determined at t-zero and at the end of temperature excursions are displayed as blue circles. Open circles were not used for kinetic modeling, but for verification. (**b**) Temperature fluctuations experienced by a vaccine during storage for 100 days, including successive temperature excursions at +20 °C and in an incubator set at +40 °C, displayed as a red line. Antigenicity predictions are displayed as blue lines, including the 95% prediction band as dashed lines. Experimental data determined at t-zero and at the end of temperature excursions are displayed as blue circles. Open circles were not used for kinetic modeling, but for verification.

## Data Availability

The data presented in this study can be available on request from the corresponding author.

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
