# Peer review of "Use of Stability Modeling to Support Accelerated Vaccine Development and Supply"

_vaccines, 2021, doi:10.3390/vaccines9101114_

Round 1

Reviewer 1 Report

This is an interesting work dealing with accelerated predictive stability by using advanced kinetic modeling. The whole work is very interesting and could have potential applications.

POINTS FOR IMPROVEMENT:

  1. Please, provide statistical analysis (trust region, etc) for the fitting made
  2. Please, include references for experimental data in the figure captions.
  3. Please, provide predictions such as a parametric analysis  at higher temperatures to further demonstrate the activity loss.
  4. Please, propose ideas for future work.
  5. A major question regarding this work is how external conditions affect loss of activity. In other words, a simple example of heat transfer by radiation or natural convention has to be examined.

In my opinion this work could be published after revision.

Author Response

Response: Thank you! We sincerely appreciated kind comments. For an accurate description of the modeling method used, kinetic parameters were explained in lines 141-143. Furthermore, additionnal theorical background is available in ref. [2-4], as already mentioned in line 148. Finally, an additional recent reference [20] from the IQ consortium and aligned with our sentence in lines 381-382 was added. 

Reviewer 2 Report

Campa and coworkers provide a method to model vaccine stability based on the Sestak-Berggren approach and Arrhenius' equation. Their model includes data of multiple temperatures in a single model. They model examples of vaccine degradation based on models of different complexity and show that data not included in the model is predicted well. Using their model, not only decay in constant temperature, but also variable temperatures, e.g. due to cold chain breaks, can be modelled. They show that data under accelerated conditions can be incorporated in the model and improve its applicability.

Author Response

Response: Thank you! We sincerely appreciated kind comments. English language and style were reviewed, leading to minor appropriate grammatical edits in the re-submitted manuscript.